# A Machine Learning Model Based on First-Trimester Lipidomic Signatures for Predicting Metabolic Pregnancy Complications

**DOI:** 10.3390/ijms262411824

**Published:** 2025-12-07

**Authors:** Alisa Tokareva, Natalia A. Frankevich, Vitaliy Chagovets, Anna Derenko, Vadim Lagutin, Vladimir Frankevich, Gennady Sukhikh

**Affiliations:** 1V.I. Kulakov National Medical Research Center for Obstetrics, Gynecology and Perinatology, Ministry of Healthcare of Russian Federation, Moscow 117997, Russia; a_tokareva@oparina4.ru (A.T.); n_frankevich@oparina4.ru (N.A.F.); v_chagovets@oparina4.ru (V.C.); a_derenko@oparina4.ru (A.D.); v_lagutin@oparina4.ru (V.L.); g_sukhikh@oparina4.ru (G.S.); 2Laboratory of Translational Medicine, Siberian State Medical University, Tomsk 634050, Russia; 3Department of Obstetrics, Gynecology, Perinatology and Reproductology, Institute of Professional Education, Federal State Autonomous Educational Institution of Higher Education, I.M. Sechenov First Moscow State Medical University of the Ministry of Health of the Russian Federation, Moscow 119991, Russia

**Keywords:** gestational diabetes mellitus, macrosomia, pregnancy, lipids, first trimester, random forest, XGBoost, mass-spectrometry

## Abstract

Gestational diabetes mellitus (GDM) and macrosomia are crucial for improving maternal and neonatal outcomes. Molecular dysregulations can manifest long before clinical symptoms appear. This study aimed to leverage first-trimester serum lipidomic signatures to build early predictive models for these complications. A case–control study was conducted using serum samples from 119 women during first-trimester screening: 40 cases and 79 controls for GDM prediction and 45 cases and 74 controls for macrosomia prediction (newborn weight more than 90 percentile). Lipidomic profiling was performed using shotgun mass spectrometry in both positive and negative electrospray ionization modes. After feature selection based on Shapley values, machine learning models—including Random Forest and XGBoost—were constructed and evaluated via 10-fold cross-validation. For GDM, potential early biomarkers included elevated levels of triacylglycerol (TG) 55:7 and decreased levels of 13-Docosenamide, plasmenyl-phosphatidylcholine (PC P)-36:2, and phosphatidylcholine (PC) 42:7. For macrosomia, phosphatidylglycerol (PG) (i-, a- 29:0), 4-Hydroxybutyric acid, and Pantothenol were significantly altered. The model for GDM prediction achieved a sensitivity of 87% and specificity of 89%. For macrosomia, the model demonstrated a sensitivity of 87% and specificity of 93%. The Random Forest and XGBoost models demonstrated comparable performance metrics on average. The risk ratio between the high- and low-risk groups defined by the models was 11.9 for GDM and 11.1 for macrosomia. Our findings demonstrate that first-trimester serum lipidomic profiles, combined with clinical data and interpreted by advanced machine learning, can accurately identify patients at high risk for GDM and macrosomia. This integrated approach holds significant promise for developing a clinical tool for timely intervention and personalized pregnancy management.

## 1. Introduction

### 1.1. The Clinical Challenge of Metabolic Pregnancy Complications

Gestational diabetes mellitus (GDM) and fetal macrosomia represent two of the most prevalent metabolic complications of pregnancy, posing significant short- and long-term risks to both maternal and child health. Current global statistics indicate that GDM develops in approximately 16% of pregnant women, with its prevalence rising in parallel with the increasing rates of obesity and type 2 diabetes [1]. Similarly, the incidence of large-for-gestational-age newborns, a proxy for macrosomia, exhibits considerable variation across Europe, ranging from 8% to 24% [2]. These conditions are not merely transient concerns of gestation. They are associated with a spectrum of adverse obstetric outcomes, including preeclampsia, gestational hypertension, polyhydramnios, preterm birth, birth trauma, and an elevated risk of perinatal mortality [3,4,5]. Beyond parturition, the long-term implications are profound. Mothers with a history of GDM face a markedly increased lifetime risk of progressing to type 2 diabetes, metabolic syndrome, and cardiovascular diseases [6,7,8]. Their offspring are predisposed to a higher likelihood of developing diabetes, pre-diabetic states, obesity, and elevated blood pressure later in life, effectively perpetuating a cycle of metabolic disease [9,10,11]. The ability to accurately identify at-risk women at the earliest possible stage is therefore a critical imperative in prenatal care, creating a window of opportunity for preventive strategies and personalized monitoring to mitigate these cascading health burdens.

### 1.2. The Search for Early Predictive Biomarkers

The current diagnostic paradigm for GDM, typically based on an oral glucose tolerance test (OGTT) at 24–28 weeks of gestation, identifies the disorder only after its metabolic manifestations are well-established. In that time, such clinical parameters as body mass index, level of fasting glucose and fasting insulin are already statistical significant altered in case of future GDM in the first trimester [12].

This reactive approach misses the crucial window for early intervention. Consequently, substantial research efforts have been directed toward discovering first-trimester biomarkers that could forecast the later development of metabolic complications. The physiological adaptations of pregnancy, including profound alterations in maternal metabolism, are reflected in systemic changes in the proteome, metabolome, and lipidome. Indeed, several studies have demonstrated that metabolic dysregulations associated with GDM can be detected in the maternal bloodstream as early as the first trimester [13,14,15]. For instance, investigations into the proteome have revealed altered levels of specific proteins involved in inflammation and insulin signaling [13,15]. Similarly, metabolomic studies have identified early shifts in amino acid and fatty acid profiles that are predictive of GDM [14,16].

### 1.3. The Lipidome as a Rich Source of Early Markers

Among the various “omics” layers, the lipidome is particularly promising. Lipids are not merely passive energy stores; they are fundamental structural components of cellular membranes and play active roles as signaling molecules, influencing insulin sensitivity, inflammation, and placental function. The lipid profile is highly dynamic during pregnancy, and its disruption can be a primary event in the pathogenesis of metabolic disorders. Supporting this, several prospective studies have already linked specific first-trimester lipidomic signatures to the subsequent risk of GDM [14,17,18]. Wang et al. and Rahman et al. demonstrated that a panel of phospholipids and triacylglycerols measured in early pregnancy could predict GDM with considerable accuracy, underscoring the diagnostic potential of the lipidome [17,18].

### 1.4. Advancing Prediction with Machine Learning and Integrated Data

While these studies establish the principle of early prediction, the integration of lipidomic data with advanced computational methods remains an area of active development. Many existing models rely on traditional statistical approaches like logistic regression [14,16,19,20]. However, machine learning (ML) algorithms, such as Random Forest and XGBoost, offer powerful advantages for handling high-dimensional, complex biological data. They can capture non-linear relationships and interactions between variables, potentially leading to more robust and accurate predictive models. Furthermore, many models focus predominantly on maternal factors. The contribution of paternal parameters, such as paternal birth weight, which may reflect genetic predispositions, has been largely overlooked, despite its potential biological relevance.

### 1.5. Study Aim and Novelty

The primary aim of this work was to develop and validate early predictive models for GDM and macrosomia by integrating first-trimester serum lipidomic profiles, acquired using high-resolution mass spectrometry, with comprehensive clinical data. We hypothesized that an integrated approach, combining detailed lipidomic profiles with comprehensive clinical data—including novel paternal parameters—and leveraging explainable machine learning, would yield a highly accurate and biologically interpretable tool for early risk stratification. This study distinguishes itself by its holistic methodology: the application of multiple ML models (Random Forest, XGBoost, MLP) with hyperparameter optimization, the use of Shapley values for model interpretation and biomarker discovery, and the inclusion of paternal medical history to create a more comprehensive predictive framework for metabolic pregnancy complications.

## 2. Results

### 2.1. Clinical Parameters

The frequency of in vitro fertilization (IVF) was statistically significantly higher in the groups with pregnancy complications compared to the control group (*p* = 0.03). Maternal birth weight in the control group (3.34 (3.15; 3.5) kg) was statistically significantly lower than in the groups with macrosomia: 3.7 (3.5; 4.1) kg, *p* = 0.001 for isolated macrosomia and 3.6 (3.5; 3.92) kg, *p* = 0.02 for macrosomia with GDM. Paternal age differed significantly across the groups (*p* = 0.04), with a trend toward being lower in the control group. Similarly, paternal birth weight in the control group (3.5 (3.3; 3.7) kg) was statistically significantly lower than in the isolated macrosomia group (3.8 (3.52; 4) kg, *p* = 0.03).

A history of GDM was statistically significantly more frequent in the group of pregnant women whose pregnancy was complicated by both GDM and macrosomia (*p* = 0.02). Endometriosis was statistically significantly more common in the groups of pregnant women whose pregnancy was complicated by macrosomia (*p* = 0.02). In the control group, a history of macrosomia (1 (2%)) was statistically significantly less frequent than in the groups with macrosomia: 8 (27%), *p* = 0.02 for isolated macrosomia and 6 (40%), *p* = 0.002 for GDM combined with macrosomia. Cesarean section in the control group (10 (20%)) was performed statistically significantly less often than in the groups with isolated GDM (15 (60%), *p* = 0.001) and isolated macrosomia (19 (63%), *p* = 0.001).

The discharge of the mother and child from the hospital in the control group (3 (3; 4) and 3 (2; 4) days, respectively) occurred statistically significantly faster than in the isolated GDM group (4 (4; 5) days and 4 (3; 5) days, *p* = 0.002) and the isolated macrosomia group (4 (4; 5) days and 4 (3; 5) days, *p* < 0.001) (Appendix A).

### 2.2. Diagnostic Models

Gradient boosting-based models demonstrated superior performance for predicting the development of GDM and macrosomia using the positive ion mode lipid profile, and for predicting GDM using the negative ion mode lipid profile (Table 1).

Five compounds from the positive ion mode—908.7800 **m*/*z** (identified as TG 55:7 + NH_4_^+^), 338.3445 **m*/*z** (13-Docosenamide + H^+^), 770.6095 **m*/*z** (PC P-36:2 + H^+^), 663.4591 **m*/*z** (PG (i-, a- 29:0) + H^+^ − H_2_O), and 860.6211 **m*/*z** (PC 42:7 + H—two compounds from the negative ion mode—299.0065 **m*/*z** and 295.2112 **m*/*z**—and three clinical parameters—maternal pre-pregnancy BMI, maternal birth weight, and history of macrosomia—were selected as potential first-trimester markers for the subsequent development of GDM during pregnancy (Appendix A; Table 2). The levels of TG 55:7 and the compound at 299.0065 **m*/*z**, along with maternal BMI, were statistically significantly higher in women who later developed GDM, whereas the levels of 13-Docosenamide, PC P-36:2, PC 42:7, and the compound at 295.2112 **m*/*z** were statistically significantly lower in the future GDM group (Table 2). In addition, 5 triacylglycerols—TG 55:7, TG 55:8, TG 58:10, TG 51:5, and TG 49:3—also had statistically significant increasing of levels in case of GDM (Appendix A). Also, levels of phosphatidylcholines PC 42:6, PC P-36:2, PC P-34:2, PC 34:1, PC 40:3, PC 40:5, PC 40:4, PC 42:7, PC 35:1, PC 38:1, PC 36:1, PC 38:2, PC 36:0, PC P-34:0, PC P-40:3, PC P-40:4, and PC P-32:0 had statistically significant decreasing in case of GDM (Appendix A). PG (a-, i- 29:0) was the only identified phosphatidylglycerol.

One compound from the positive ion mode—663.4591 **m*/*z** (PG (i-, a- 29:0) + H^+^ − H_2_O)^+^—and nine compounds from the negative ion mode—165.0317 **m*/*z** (4-Hydroxybutyric acid + HCO_3_^−^)^−^, 234.1434 **m*/*z**, 174.9463 **m*/*z**, 239.1149 **m*/*z**, 951.1787 **m*/*z**, 250.1309 **m*/*z** (Pantothenol + HCO_2_^−^)^−^, 247.1564 **m*/*z**, 374.2242 **m*/*z**, and 195.1282 **m*/*z**—along with four clinical parameters (maternal pre-pregnancy BMI, maternal birth weight, history of macrosomia, and paternal birth weight) were selected as potential first-trimester markers for the subsequent development of macrosomia during pregnancy (Appendix A; Table 2). The levels of PG (i-, a- 29:0), 4-Hydroxybutyric acid, and the compounds at **m*/*z** 234.1434, 239.1149, 951.1787, 247.1564, 374.2242, and 195.1282, along with Pantothenol, maternal pre-pregnancy BMI, paternal birth weight, and the frequency of macrosomia in the medical history, were statistically significantly higher. In contrast, the level of the compound at 174.9463 **m*/*z** was statistically significantly lower in the group that subsequently developed macrosomia (Table 2).

Finally, identified lipid markers are included in phosphatidylcholines (PC P-36:2, PC 42:7), triacylglycerols (TG 55:7), phosphatidylglycerols (PG (i-, a- 29:0)), and fatty acyls (13-Docosenamide, 4-Hydroxybutyric acid, and Pantothenol).

The best predictive performance for GDM was demonstrated by the XGBoost-based model, with a sensitivity of 87%, specificity of 89%, and accuracy of 88%. For macrosomia, the Random Forest-based model showed the best performance, with a sensitivity of 87%, specificity of 93%, and accuracy of 91% (Figure 1; Table 3).

Thus, we developed models capable of identifying patients belonging to the high-risk group for developing GDM and the high-risk group for developing macrosomia, with risk ratios of 11.9 (10.3–13.9) and 11.1 (10.0–12.3), respectively.

## 3. Discussion

Our study demonstrates that integrating first-trimester lipidomic signatures from high-resolution mass spectrometry with paternal and maternal clinical parameters enables the highly accurate prediction of GDM and macrosomia. Beyond the predictive models, a key finding is the identification of a panel of specific lipid species, including TG 55:7, PC P-36:2, and 4-Hydroxybutyric acid, which are significantly dysregulated weeks before clinical diagnosis. These markers point to early disruptions in specific metabolic pathways, offering novel insights into the underlying pathophysiology.

Yang et al. [14] demonstrated the efficacy of a logistic regression model that utilized maternal age, pre-pregnancy BMI, race, family history of diabetes, and a blood profile for prediction. This blood profile included, alongside the proteome and glucose, information on the levels of even- and odd-chain short-chain fatty acids. This model achieved an area under the receiver operating characteristic curve (AUC) of 0.84 [14]. A logistic regression model presented by Wang et al., which used a combination of clinical parameters and lipid signatures, also showed diagnostic potential (AUC = 0.80) [17]. A model of comparable performance was demonstrated by Manna et al., where a logistic regression model was trained on maternal clinical parameters and blood metabolomic parameters (blood fatty acid profile and amino acid profile), yielding an AUC of 0.84, a false positive rate of 10%, and a detection rate of 60% [16]. It is noteworthy that for the combined model proposed in our current study, the corresponding metrics were an AUC of 0.88, a false positive rate of 11%, and a detection rate of 87%.

In contrast, a model utilizing medical history parameters based on logistic regression in the study by Tranidou et al. (2024) showed inferior predictive performance (AUC = 0.68, false positive rate 10%, detection rate 20%) compared to the Random Forest model in our current work (AUC = 0.76, false positive rate 8%, detection rate 51%) [21].

The application of advanced machine learning methods (decision trees) enabled Koos et al. to build a model using urinary metabolite levels with an accuracy of 96.7% [22].

Monari et al. proposed a logistic regression-based model for predicting fetal macrosomia using information on parity, pre-pregnancy BMI, and PAPP-A concentration in MoM, with a sensitivity of 55% and specificity of 79% [19]. This makes its performance comparable to the clinical data-based Random Forest model presented in our article (sensitivity 40%, specificity 93%). The model developed by Du et al., based on logistic regression and utilizing information on BMI, parity, history of macrosomia and GDM, as well as levels of glycated hemoglobin and total cholesterol at the first screening, demonstrated better performance: AUC = 0.81, sensitivity 71%, and specificity 78% [20]. Employing advanced machine learning methods, such as Random Forest, allowed Zhong et al. to create an integrated model using information on the abundance of four marker bacterial species in the gut microbiota and first-trimester clinical parameters (BMI, waist circumference, blood levels of albumin, total triglycerides, and total cholesterol), which showed high predictive power with an AUC of 0.91 on the test dataset [23] (Table 4).

It is important to note that, unlike the aforementioned studies, parameters such as paternal and maternal birth weights are more relevant in our study than widely used model parameters such as PAPP-A levels, beta-hCG, and first-trimester ultrasound results. In our cohort, the association between paternal birth weight with offspring birth weight is significant and influences both low-birth-weight and high-birth-weight outcomes [24,25].

The level of TG 55:7 was statistically significantly elevated in the group of women who subsequently developed GDM. This triacylglycerol has also been previously identified as being associated with nutritional imbalance [26]. The elevated level of TG 55:7, a highly unsaturated triglyceride, may reflect early alterations in hepatic lipogenesis and lipid storage driven by emerging insulin resistance. Such specific TG species are increasingly recognized as more sensitive markers of metabolic health than total triglycerides.

The level of 13-Docosenamide was statistically significantly decreased in the group of women with future GDM. Rodrigues et al. suggested that a range of metabolites, including 13-Docosenamide, possess anti-diabetic properties, based on the effect of Naregamia alata extract on diabetes symptoms in rats [27].

The level of PC P-36:2 was statistically significantly lower in the group of women who later developed GDM. Consistently, Wu et al. reported that a higher blood level of PC P-36:2 was associated with a lower risk of GDM [28]. Furthermore, Bagheri et al. and Pang S.-J. et al. found that plasmenyl phosphatidylcholine is negatively correlated with the level of insulin resistance [29,30]. Our finding that PC P-36:2 is significantly lower in women who later developed GDM is highly consistent with the known biology. Plasmalogens are endogenous antioxidants. Their depletion has been linked to insulin resistance and is a hallmark of metabolic syndrome [29,30]. The observed reduction in the first trimester suggests that an impaired antioxidant capacity and increased oxidative stress may be an early event in the pathogenesis of GDM, preceding its clinical manifestation A lower level of PC P-36:2 is also observed in women after delivery who had developed GDM during pregnancy compared to those without this complication, and a lower level of this lipid is characteristic of a higher risk of developing diabetes mellitus [30]. Similarly, a negative correlation with the level of insulin resistance has been reported for PC 42:7 [30].

Our study has several strengths, including the use of high-resolution lipidomics, the integration of novel paternal parameters, and the application of explainable AI (SHAP) for biomarker discovery. However, certain limitations should be acknowledged. The case–control design and single-center nature of our cohort, while ideal for initial discovery, necessitate external validation in a large, prospective, multi-center study to confirm generalizability. Furthermore, while we identified several lipid markers with high accuracy, the definitive structural confirmation for some signals, particularly in negative ion mode, requires further MS/MS analysis. Potential confounding factors, such as detailed dietary information, could not be fully accounted for and should be considered in future research.

In conclusion, our research moves beyond prediction by providing a molecular window into the early stages of metabolic complications in pregnancy. The identified lipid panel offers a powerful tool for risk stratification and opens new avenues for investigating the pathophysiology of GDM and macrosomia. Future work will focus on the external validation of these models and the functional characterization of the highlighted lipid pathways to uncover their precise role in pregnancy metabolism. In addition, animal studies can give more information about the early stages of metabolomic disorders during pregnancy.

## 4. Materials and Methods

### 4.1. Study Design

A case–control study was conducted at the V.I. Kulakov National Medical Research Center for Obstetrics, Gynecology, and Perinatology. Out of 1200 women who underwent first-trimester prenatal screening (11–13.6 weeks) including blood sampling, 119 patients were enrolled in the study (Appendix A). The inclusion criteria were as follows:Singleton pregnancy;Neonatal birth weight ≥ 2500 g;Absence of malignant diseases in the mother;No history of organ transplantation in the mother;Absence of pregestational type 1 or type 2 diabetes mellitus in the mother;Undergoing an oral glucose tolerance test at 24–28 weeks of gestation and delivery at the center;Absence of congenital malformations in the mother and fetus;Absence of other major pregnancy complications;Provision of informed consent by the mother for participation in the study.

Diagnosis of GDM was based on an oral glucose tolerance test (OGTT) performed after an 8–14 h fast. Glucose levels were measured at fasting, and a diagnosis of GDM was made if the fasting glucose level exceeded 5.1 mmol/L. If the fasting level was normal, the pregnant woman received a load of 75 g of glucose dissolved in 200–300 mL of water. Blood glucose levels were then measured 1 h and 2 h after the glucose load. A diagnosis of GDM was made if the 1 h glucose level was above 10.0 mmol/L or the 2 h level was above 8.5 mmol/L. Diagnosis of macrosomia was made when the neonatal birth weight exceeded the 90th percentile.

The study groups were formed as follows: the subgroup without macrosomia and without GDM (Group 1, control) included 49 women; the subgroup with isolated GDM (Group 2) included 25 women; the subgroup with isolated macrosomia (Group 3) included 30 women; and the subgroup with a combination of GDM and macrosomia (Group 4) included 15 women. Among the pregnant women in the isolated GDM group (Group 2), 14 received dietary therapy and 6 required insulin therapy. In the group with combined GDM and macrosomia (Group 4), 5 women received dietary therapy and 4 required insulin therapy.

### 4.2. Sample Collection and Preparation

Blood samples were collected by venipuncture into sterile 9 mL vacuum tubes containing separation gel, following a 12 h fasting period. The collected serum was centrifuged at 700× *g* for 10 min at 4 °C. The supernatant was transferred into sterile tubes and stored at −80 °C until further analysis. Lipid extraction was performed using a modified Folch method [31]. The collected organic layer was dried under a stream of nitrogen and reconstituted in an acetonitrile/isopropanol mixture (1:1, *v*/*v*) for subsequent mass spectrometric analysis.

### 4.3. Lipidomic Mass Spectrometric Analysis

Mass spectrometric analysis was performed using direct infusion electrospray ionization on a Maxis Impact qTOF mass spectrometer (Bruker Daltonics, Bremen, Germany) [32]. The analysis was conducted in both positive and negative ion modes across a mass range of 100–1800 *m*/*z*, with the following parameters: capillary voltage of 4.1 kV in positive mode and 3.0 kV in negative mode, nebulizer gas pressure of 0.7 bar, and drying gas flow rate and temperature of 6 L/min and 200 °C, respectively. A 20 μL aliquot of the sample was introduced into a methanol/water (9:1, *v*/*v*) mobile phase flowing at 10 μL/min using a Dionex UltiMate 3000 system (ThermoScientific, Bremen, Germany).

Following mass spectrometric analysis, 100 mass spectra acquired during sample elution were averaged, aligned by total ion current, and processed into a data matrix containing the intensity of each peak with a specific *m*/*z* value in each sample.

### 4.4. Statistical Analysis

Comparison of clinical parameters between groups was performed using the Kruskal–Wallis test for numerical variables and Pearson’s chi-square test for categorical variables. Differences were considered statistically significant at *p* < 0.05. For parameters that showed statistically significant differences, pairwise comparisons were conducted using Dunn’s test for numerical variables and pairwise chi-square tests for categorical variables, with statistical significance set at *p* < 0.05.

Mass spectrometric profiles were checked for feature collinearity using Pearson’s correlation test. Features with a correlation coefficient greater than 0.9 were excluded, retaining the one with the lower mass-to-charge ratio.

For the positive ion mode mass spectrometric profiles, negative ion mode profiles, and the set of clinical parameters—which included paternal and maternal age, paternal and maternal birth weight, maternal pre-pregnancy BMI, family history of diabetes mellitus, history of GDM and macrosomia in previous pregnancies, parity, gravidity, number of deliveries, primiparity status, primigravida status, medical history, first-trimester levels of ß-hCG and PAPP-A, weight gain by the first screening, crown-rump length, biparietal diameter, head circumference, abdominal circumference, nuchal translucency thickness, placental thickness on first-trimester ultrasound, and fetal sex—predictive models for GDM and macrosomia were built using Random Forest [33], XGBoost (Extreme Gradient Boosting) [34], and Multilayer Perceptron (MLP). Hyperparameters for the XGBoost and MLP models were optimized using the Particle Swarm Optimization method [35].

For each classification task (predicting GDM and macrosomia) and each dataset, the optimal model was selected from the three candidates (Random Forest, XGBoost, and MLP) based on maximizing accuracy, sensitivity, and specificity, as determined by 10-fold cross-validation. For the variables incorporated into each optimal model, Shapley values were computed [36]. Features with Shapley values no less than half of the maximum value were considered potential markers. Potential lipid markers were identified with a mass accuracy tolerance of <0.01 Da using the Human Metabolome Database [37], searching for lipids and lipid-like compounds in blood while excluding compounds of specific exogenous origin. Subsequently, models based on the potential markers for GDM/macrosomia were built using the previously selected optimal method. For the final model, accuracy, sensitivity, specificity, and positive and negative predictive values were calculated.

The analysis was performed using scripts in R version 4.3.3 (Vienna, Austria) [38] with the following packages: caret 7.0-1 [39], xgboost 1.7.8.1 [34], keras 2.15.0 [40], randomForest 4.7-1.2 [33], kernelshap 0.7.0 [41], ranger 0.17.0 [42], pROC 1.18.5 [43], shapviz 0.9.7 [44], and ggplot2 3.5.2 [45].

## 5. Conclusions

In this study, we successfully developed and internally validated robust machine learning models capable of accurately predicting the risk of gestational diabetes mellitus (GDM) and macrosomia as early as the first trimester of pregnancy. Integrating lipid markers with standard clinical parameters improved the predictive power of the models, highlighting the unique contribution of first-trimester lipidomic dysregulation.

A key scientific contribution of this work is the identification of a panel of specific lipid species, including TG 55:7, PC P-36:2, and PG (i-, a- 29:0), whose altered levels signify early metabolic dysregulation preceding clinical diagnosis. The association of these lipids with pathways such as insulin resistance and nutritional imbalance provides a plausible molecular basis for the pathogenesis of these complications and offers new targets for mechanistic investigation.

The implementation of such a predictive tool in clinical practice could improve prenatal care by enabling an early implementation of preventive action, such as tailored dietary plans for high-risk patients.

However, to advance these findings towards clinical application, several steps are necessary. Future research must focus on the external validation of these models in large, multi-center, prospective cohorts to ensure generalizability. Furthermore, a critical priority is the definitive structural identification of the unknown lipid signatures, particularly those detected in negative ion mode, using MS/MS fragmentation to fully elucidate their biological roles and diagnostic utility. A cost-effectiveness analysis compared to existing diagnostic methods is necessary. Furthermore, the most user-friendly implementation of this model would be a software tool, such as a standalone application or a web service.

## Figures and Tables

**Figure 1 ijms-26-11824-f001:**
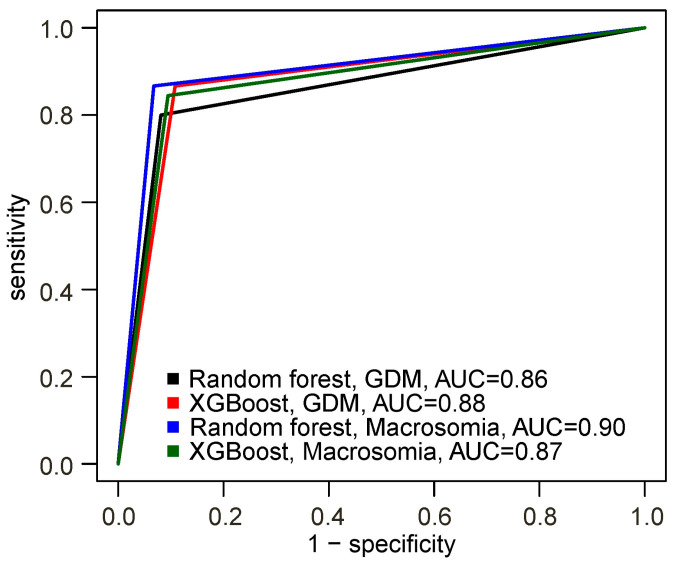
Receiver operating characteristic (ROC) curves of the models built using the selected markers.

**Table 1 ijms-26-11824-t001:** Performance metrics of models created using the full datasets. The best-performing models for each data type and outcome are highlighted in **bold**.

Data Type	Method	Outcome	Sens.	Spec.	Acc.
**Lipids, Positive Ion Mode**					
	Random Forest	GDM	0.23	0.95	0.71
	**XGBoost**	**GDM**	**0.60**	**0.89**	**0.79**
	MLP	GDM	0.03	0.99	0.66
	Random Forest	Macrosomia	0.44	0.95	0.76
	**XGBoost**	**Macrosomia**	**0.76**	**0.80**	**0.78**
	MLP	Macrosomia	0.56	0.73	0.66
**Lipids, Negative Ion Mode**					
	Random Forest	GDM	0.20	0.92	0.68
	**XGBoost**	**GDM**	**0.53**	**0.80**	**0.71**
	MLP	GDM	0.83	0.37	0.52
	**Random Forest**	**Macrosomia**	**0.84**	**0.95**	**0.91**
	XGBoost	Macrosomia	0.64	0.84	0.76
	MLP	Macrosomia	0.56	0.73	0.66
**Clinical Parameters**					
	**Random Forest**	**GDM**	**0.51**	**0.92**	**0.76**
	XGBoost	GDM	0.47	0.77	0.65
	MLP	GDM	0.43	0.68	0.60
	**Random Forest**	**Macrosomia**	**0.40**	**0.93**	**0.73**
	XGBoost	Macrosomia	0.91	0.09	0.40
	MLP	Macrosomia	0.67	0.35	0.47

**Table 2 ijms-26-11824-t002:** Potential first-trimester markers for the subsequent development of pregnancy complications. For lipid markers, the potential identification with lipid class in brackets, ion, absolute difference between measured and theoretical mass-to-charge ratio (Δ*m*/*z*), theoretical and measured *m*/*z*, *p*-value (Mann–Whitney test, non-adjusted and after Benjamini–Hochberg correction (FDR)), and the ratio of medians (with/without complication) are provided. For quantitative clinical parameters, *p*-value (Mann–Whitney test, non-adjusted and after Benjamini–Hochberg correction (FDR)), and the ratio of medians are provided. For categorical clinical parameters, and the risk ratio are provided. M—neutral molecule.

Marker	Ion	|Δ*m*/*z*|	Theoretical *m*/*z*	Measured *m*/*z*	*p*-Value (FDR)	MR/HR
**Lipids, Positive Ion Mode, GDM**						
TG 55:7 (triacylglycerol)	(M + NH_4_^+^)^+^	0.009	908.7710	908.7800	<0.001 (0.003)	1.78
13-Docosenamide (fatty acyl)	(M + H^+^)^+^	0.002	338.3420	338.3445	0.01 (0.08)	0.73
PC P-36:2 (phosphatidylcholine)	(M + H^+^)^+^	0.003	770.606	770.6095	<0.001 (0.01)	0.80
PG (i-, a- 29:0) (phosphatidylglycerol)	(M + H^+^-H_2_O)^+^	0.001	663.46	663.4591	0.76 (0.88)	0.89
PC 42:7 (phosphatidylcholine)	(M+H^+^)^+^	0.004	860.6170	860.6211	0.02 (0.11)	0.91
**Lipids, Negative Ion Mode, GDM**						
299.0065	-	-	-	299.0065	<0.001 (0.003)	>10
295.2112	-	-	-	295.2112	0.001 (0.04)	0.88
**Clinical Parameters, GDM**						
Maternal BMI, kg/m^2^					0.003 (0.12)	1.10
Maternal birth weight, kg					0.55 (1)	1.00
Macrosomia in history, n (%)					0.52 (1)	1.40
**Lipids, Positive Ion Mode, Macrosomia**						
PG (i-, a- 29:0) (phosphatidylglycerol)	(M+H^+^ − H_2_O)^+^	0.001	663.4600	663.4591	<0.001 (<0.001)	3.44
**Lipids, Negative Ion Mode, Macrosomia**						
4-Hydroxybutyric acid (fatty acyls)	(M+HCO_3_^−^)^−^	0.008	165.0400	165.0317	<0.001 (<0.001)	3.33
234.1434	-	-	-	234.1434	<0.001 (<0.001)	3.22
174.9463	-	-	-	174.9463	<0.001 (<0.001)	0.84
239.1149	-	-	-	239.1149	<0.001 (<0.001)	>10
951.1787	-	-	-	951.1787	<0.001 (<0.001)	>10
Pantothenol (fatty acyls)	(M+HCO_2_^−^)^−^	0.002	250.129	250.1309	<0.001 (<0.001)	2.55
247.1564	-	-	-	247.1564	<0.001 (<0.001)	2.38
374.2242	-	-	-	374.2242	<0.001 (<0.001)	2.01
195.1282	-	-	-	195.1282	<0.001 (<0.001)	1.45
**Clinical Parameters, Macrosomia**						
Maternal BMI, kg/m^2^					<0.001 (0.003)	1.13
Maternal birth weight, kg					<0.001 (0.002)	1.06
Macrosomia in history, *n* (%)					<0.001 (0.001)	2.80
Paternal birth weight, kg					0.004 (0.04)	1.12

**Table 3 ijms-26-11824-t003:** Performance metrics of the final models for predicting pregnancy complications.

Complication	Model	Sensitivity	Specificity	Accuracy	AUC	PPV	NPV	F-Score
**GDM**	XGBoost	0.87	0.89	0.88	0.88	0.80	0.93	0.83
**Macrosomia**	Random Forest	0.87	0.93	0.91	0.90	0.89	0.92	0.89

**Table 4 ijms-26-11824-t004:** Quality metrics of the diagnostic models.

Authors, Year	Reference	Complication	Method	Features	AUC	Sensitivity	Specificity
Yang et al., 2025	[14]	GDM	Logistic regression	Routine clinical parameters, proteome, metabolome biomarker	0.84	75%	75%
Wang, et al., 2021	[17]	GDM	Logistic regression	Routine clinical parameters, metabolome biomarkers	0.80	~75%	~75%
Manna, et al., 2025	[16]	GDM	Logistic regression	Metabolome biomarkers	0.84	60%	90%
Tranidou et al., 2024	[21]	GDM	Logistic regression	Medical history and routine screening pregnancy markers	0.68	20%	90%
Koos et al., 2021	[22]	GDM	Decision tree	Urinary metabolome	0.99	97.8%	95.7%
Monari et al., 2021	[19]	Macrosomia	Logistic regression	Routine clinical parameters	0.705	55.2%	79.0%
Du et al., 2022	[20]	Macrosomia	Logistic regression	Routine clinical parameters	0.807	71.6%	77.7%
Zhong Z et al., 2024	[23]	Macrosomia	Random forest	Routine screening parameters, gut microbiota species	0.91	85.71%	81.82%

## Data Availability

The original contributions presented in this study are included in the article. Further inquiries can be directed at the corresponding author.

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
