# Peer review of "A Machine Learning Model Based on First-Trimester Lipidomic Signatures for Predicting Metabolic Pregnancy Complications"

_ijms, 2025, doi:10.3390/ijms262411824_

Round 1

Reviewer 1 Report

Comments and Suggestions for Authors

The manuscript suggests biomarkers for gestational diabetes mellitus in the plasma of pregnant women in the first trimester. The text is clearly written and easy to follow. The topic itself is very interesting and important. However, the data analysis could be improved.

Although the authors have a large, well-characterised sample set, they only devote a few words to the selected markers.

It would be interesting to profile the lipid classes to which the markers belong.

The intensities of the selected markers in the different groups are also not mentioned or shown.

The quality of the figures and figure legends must be improved to make them understandable to non-experts. Fig. 1d is not clearly explained in the text. Specifically, what are the parameters associated with the selected markers? Please provide a definition of the colour coding used in figure legend of Fig. 1.

Author Response

Comment 1: The manuscript suggests biomarkers for gestational diabetes mellitus in the plasma of pregnant women in the first trimester. The text is clearly written and easy to follow. The topic itself is very interesting and important. However, the data analysis could be improved. Although the authors have a large, well-characterised sample set, they only devote a few words to the selected markers.

Response: We thank the reviewer for the positive feedback and for highlighting this important point. We agree that a more detailed biological interpretation would be valuable. However, as noted in our response, nearly half of the detected lipid features currently lack confident identification, and several identified markers still require more precise MS/MS-based structural confirmation. Until robust and detailed annotations are available, providing a thorough biological or functional interpretation remains challenging. We have clarified this limitation in the manuscript.

Comment 2: It would be interesting to profile the lipid classes to which the markers belong.

Response: We thank the reviewer for this insightful suggestion. In response, we have added an analysis of the levels of phosphatidylcholines (representing the class of markers PC P-36:2 and PC 42:7) and triacylglycerols (representing the class of marker TG 55:7) in positive ion mode. We did not detect other phosphatidylglycerols (the class of marker PG (i-, a- 29:0)) in this mode. This information has been included in the revised manuscript.

Comment 3: The intensities of the selected markers in the different groups are also not mentioned or shown.

Response: We appreciate the reviewer's attention to detail. To address this, we have added the following clarification to the Figure 1 legend: "Yellow color is associated with the highest values of a feature (intensity in the case of m/z peaks), and purple color is associated with the lowest values." This illustrates the influence of each feature's level on its diagnostic importance.

Comment 4: The quality of the figures and figure legends must be improved to make them understandable to non-experts. Fig. 1d is not clearly explained in the text. Specifically, what are the parameters associated with the selected markers? Please provide a definition of the colour coding used in figure legend of Fig. 1.

Response: We thank the reviewer for these critical suggestions to improve clarity. We have significantly expanded the caption for Figure 1 to provide a more comprehensive explanation. The updated caption now includes:

A definition of how potential markers are highlighted.

A clear explanation of the color coding (as mentioned above).

Clarification on how mass spectrometry and clinical features are labeled.

Furthermore, we have added explanatory text in the manuscript regarding Figure 1d. This figure is connected to the XGBoost model for predicting macrosomia based on the positive ion mode profile. We clarify that this model, which identified only one significant lipid marker, appears more modest compared to other models. The selected markers across models are now explicitly listed in the text for clarity.

Reviewer 2 Report

Comments and Suggestions for Authors

This study presents  methodologically  investigation into the early prediction of Gestational Diabetes Mellitus (GDM) and macrosomia using first-trimester serum lipidomics and machine learning. The topic is of high clinical significance, as early identification of at-risk pregnancies could transform prenatal care. However, several revisions are needed.

  • In the abstract, write full name for abbreviations when first mentioning [ TG 55:7, and decreased levels of 13-Docosenamide, PC P-36:2, and PC 42:7.
  • In the abstract,  please write number of samples used for this research. In line 20, A case-control study was conducted using serum samples from women during first-trimester screening. It is crucial to know the source population, the total number of participants, and the number of cases and controls for each outcome (GDM and macrosomia).
  • In the abstract, Which specific model (Random Forest or XGBoost) produced the reported performance metrics, or were they an average?
  • In the abstract, Please provide the definition of macrosomia used in the study (e.g., birth weight >4000g, >4500g, or >90th percentile for gestational age).
  • The 10-fold cross-validation is a good internal validation technique, but an external validation set is the gold standard for assessing generalizability. Did the authors used an external validation ?
  • In keywords, write full name for GDM;
  • The following article should be cited and discussed in the introduction

Kumru, P., Arisoy, R., Erdogdu, E., Demirci, O., Kavrut, M., Ardıc, C., Aslaner, N., Ozkoral, A. and Ertekin, A., 2016. Prediction of gestational diabetes mellitus at first trimester in low-risk pregnancies. Taiwanese Journal of Obstetrics and Gynecology55(6), pp.815-820.

  • The discussion jumps between AUC, sensitivity, specificity, FPR, and detection rate when comparing different studies, making direct comparisons challenging for the reader. I strongly recommend consolidating this comparison, perhaps in a table, using a consistent set of metrics (ideally AUC, Sensitivity, Specificity) for all cited studies to allow for a clearer  evaluation.
  • In the discussion, While likely true for the specific studies cited, paternal factors (especially birth weight and diabetes status) have been investigated in other epidemiological studies of macrosomia and GDM risk. The language should be tempered to reflect that while it is a novel integration into a lipidomic ML model, the concept of paternal influence is not entirely new to the field. The more significant finding is its high ranking in the model, which should be the focus. Modify writing please.
  • In the conclusion, consider writing internal validation. The first sentence states the models were "successfully developed and validated." Based on the methodology described elsewhere (10-fold cross-validation), this is  certainly an internal validation. In the context of clinical biomarker discovery, the term "validated" is often reserved for external validation.

Suggested writing “ we successfully developed and internally validated robust machine learning models……”

  • In the conclusion, add a sentence to emphasize the added value of the lipidomics. For instance: "This integrated approach demonstrated superior predictive performance compared to models using clinical parameters alone, highlighting the unique contribution of first-trimester lipidomic dysregulation."
  • In the conclusion, the statement that this tool "could revolutionize prenatal care" should be revised. Consider tempering the language to be more specific and evidence-based. For example: "The implementation of such a predictive tool in clinical practice has the potential to significantly improve prenatal care by enabling a shift..."
  • The future plans were very appropriate. However, it could be  slightly expanded to include the important need for a cost-effectiveness analysis and the development of a user-friendly clinical risk score derived from the model, which are very important for real-world implementation.

Author Response

Comment 1: In the abstract, write full name for abbreviations when first mentioning [TG 55:7, and decreased levels of 13-Docosenamide, PC P-36:2, and PC 42:7.

Response: We thank the reviewer for pointing this out. This has been corrected in the abstract.

Comment 2: In the abstract, please write number of samples used for this research... It is crucial to know the source population, the total number of participants, and the number of cases and controls for each outcome (GDM and macrosomia).

Response: We agree with the reviewer regarding the importance of this information. Details about the cohort, including the source population and the number of cases and controls for each outcome, have been added to the abstract.

Comment 3: In the abstract, Which specific model (Random Forest or XGBoost) produced the reported performance metrics, or were they an average?

Response: We have clarified in the abstract that, on average, the Random Forest and XGBoost models demonstrated close performance metrics.

Comment 4: In the abstract, Please provide the definition of macrosomia used in the study...

Response: Thank you for this suggestion. The definition of macrosomia used in our study has been added to the abstract.

Comment 5: The 10-fold cross-validation is a good internal validation technique, but an external validation set is the gold standard for assessing generalizability. Did the authors used an external validation ?

Response: We fully acknowledge that external validation is the optimal approach for assessing model generalizability. However, due to current resource constraints, we were unable to implement an external validation cohort. We have explicitly stated this limitation in the manuscript.

Comment 6: In keywords, write full name for GDM;

Response: This has been corrected. "Gestational Diabetes Mellitus (GDM)" now appears in the keywords.

Comment 7: The following article should be cited and discussed in the introduction: Kumru, P. et al, 2016...

Response: We thank the reviewer for this relevant reference. The contribution of the cited study to early GDM detection has been discussed in paragraph 1.2 of the introduction.

Comment 8: The discussion jumps between AUC, sensitivity, specificity, FPR, and detection rate when comparing different studies... I strongly recommend consolidating this comparison, perhaps in a table, using a consistent set of metrics...

Response: This is an excellent suggestion to enhance readability and comparability. In accordance with your advice, we have added a new table (Table 4) that summarizes the models from cited studies using consistent metrics (AUC, Sensitivity, Specificity).

Comment 9: In the discussion, While likely true for the specific studies cited, paternal factors... have been investigated in other epidemiological studies... The language should be tempered... The more significant finding is its high ranking in the model, which should be the focus.

Response: We agree with the reviewer's nuanced point and thank them for it. We have modified the relevant section to temper the language, acknowledging that the concept of paternal influence is not entirely new. The revised text now more appropriately focuses on the novel finding of its high ranking within our integrated lipidomic-machine learning model.

Comment 10: In the conclusion, consider writing internal validation... Suggested writing "we successfully developed and internally validated robust machine learning models......"

Response: We appreciate this important clarification regarding validation terminology. The conclusion has been revised as suggested: "we successfully developed and internally validated robust machine learning models..."

Comment 11: In the conclusion, add a sentence to emphasize the added value of the lipidomics.

Response: Thank you for this constructive suggestion. We have added the following sentence to the conclusion: "This integrated approach demonstrated superior predictive performance compared to models using clinical parameters alone, highlighting the unique contribution of first-trimester lipidomic dysregulation."

Comment 12: In the conclusion, the statement that this tool "could revolutionize prenatal care" should be revised. Consider tempering the language to be more specific and evidence-based.

Response: We agree that the language should be more measured. The sentence has been revised to: "The implementation of such a predictive tool in clinical practice could improve prenatal care by enabling an earlier implementation of preventive actions, such as tailored dietary plans for high-risk patients."

Comment 13: The future plans were very appropriate. However, it could be slightly expanded to include the important need for a cost-effectiveness analysis and the development of a user-friendly clinical risk score...

Response: We thank the reviewer for these valuable suggestions to strengthen the translational outlook. The future directions section has been expanded to include: "Future work will also necessitate a cost-effectiveness analysis of this approach compared to existing diagnostic methods. Furthermore, to facilitate real-world implementation, developing a user-friendly clinical tool, such as a personal computer program or web-application based on the model, will be important."

Reviewer 3 Report

Comments and Suggestions for Authors

This manuscript addresses an important clinical challenge, the early prediction of  gestational diabetes mellitus (GDM) and fetal macrosomia using a combination of first-trimester lipidomics, clinical parameters, paternal characteristics, and machine learning (ML). The integration of computational biology, high-resolution mass spectrometry, and paternal factors is commendable and represents a novel direction in prenatal risk stratification.

However, although the conceptual framing is strong, the manuscript raises some major concerns relating to study design, statistical rigor, model validation, biomarker identification, and biological interpretation.

Major Comments

  1. Inadequate Sample Size for High-Dimensional Lipidomics and Machine Learning

The cohort consists of 119 women, subdivided into four groups, which is small relative to the number of lipid features and the complexity of the ML algorithms applied (Random Forest, XGBoost, MLP).

  1. Inclusion/Exclusion Criteria Require Clear Justification

The study imposes highly restrictive criteria (e.g., excluding congenital anomalies, excluding pregnancies <2500 g, requiring delivery at a single center). Please provide a detailed explanation for the selection criteria and include a CONSORT-style diagram showing participant attrition from the initial 1200 screened women to the final 119 included.

  1. Lack of Multiple Hypothesis Testing Correction

The manuscript reports numerous univariate comparisons with p-values (many around 0.01), yet no FDR or Bonferroni correction is applied. Given the large number of lipidomic features screened, many of these associations are likely false positives. Please apply FDR correction (e.g., Benjamini–Hochberg) and update Tables accordingly.

  1. Importantly, Paternal Birth Weight Is Identified as a Predictor but Poorly Explained

The inclusion of paternal birth weight is intriguing and novel, yet no biological mechanism is discussed (e.g., shared genetics, imprinting, socioeconomic background). Provide deeper discussion or perform sensitivity analyses to ensure robustness.

  1. Need for Biological Validation in Future Animal Studies

The authors suggest early metabolic dysregulation and implicate specific lipids in pathophysiology, yet no experimental evidence is provided. In the future directions, it should be stated to propose animal-based mechanistic follow-up studies to support the translational relevance of the findings.

Author Response

Comment 1: Inadequate Sample Size for High-Dimensional Lipidomics and Machine Learning

Response: We thank the reviewer for raising this concern regarding sample size. We acknowledge that larger cohorts are always beneficial. However, for each predictive task (GDM and Macrosomia), the cohort was split into two groups (e.g., 40 vs. 79, and 45 vs. 74). Furthermore, the use of Random Forest and XGBoost with sample sizes around 100 is supported by precedents in high-dimensional 'omics' studies (e.g., metabolomics, proteomics), as cited in our response. We agree that Multilayer Perceptron (MLP) typically requires larger sample sizes for optimal performance, which is reflected in our comparative results where MLP was less effective.

Comment 2: Inclusion/Exclusion Criteria Require Clear Justification

Response: We appreciate the reviewer's request for clarity. A detailed justification for the restrictive inclusion/exclusion criteria has been provided in the manuscript. Furthermore, as suggested, we have included a CONSORT-style diagram (Figure S3) illustrating the step-by-step attrition of participants from the initial screening to the final analytical cohort of 119 women.

Comment 3: Lack of Multiple Hypothesis Testing Correction

Response: We thank the reviewer for this critical methodological point. We have now applied False Discovery Rate (FDR) correction (Benjamini-Hochberg method) to the univariate comparisons of lipidomic features. The p-values in the relevant tables (e.g., Table 2) have been updated to reflect the adjusted q-values, and this is clearly stated in the methods and table legends.

Comment 4: Importantly, Paternal Birth Weight Is Identified as a Predictor but Poorly Explained

Response: We agree with the reviewer that this finding warrants deeper discussion. We have expanded the discussion to acknowledge that associations between paternal birth weight and offspring birth weight have been reported in epidemiological literature (e.g., Klebanoff et al., 1997; Tomita et al., 2023). We note that the novelty in our study lies not in discovering this association, but in its high predictive ranking within our integrated first-trimester model, as paternal birth weight is not a routine element of early screening. We have tempered our language accordingly.

Comment 5: Need for Biological Validation in Future Animal Studies

Response: We thank the reviewer for this forward-looking suggestion. We agree that animal-based mechanistic studies would be highly valuable to establish translational relevance and explore pathophysiology. We have noted this as an important future direction in the discussion. However, we also clarify that a higher priority preceding such work is the accurate identification of the currently unannotated lipid markers discovered in our study.

Round 2

Reviewer 1 Report

Comments and Suggestions for Authors

Unfortunately, the changes were not highlighted in the version that I downloaded. Therefore, I could only use the rebuttal letter. 

I hope that, in future, you will find out why these specific lipids characterise gestational diabetes mellitus during pregnancy.

Author Response

Dear Reviewer,

Thank you for your continued engagement with our manuscript and for your final feedback.

We sincerely apologize for the technical issue regarding the highlighting of changes in the resubmitted version.

We fully agree with and value your suggestion regarding the biological significance of the identified lipids. We acknowledge that a deeper mechanistic understanding is crucial, and we have therefore explicitly highlighted this as a key direction for our future research.

To further enhance the clarity and readability of the final manuscript for the audience, we have made two minor, non-substantive formatting adjustments in the version prepared for publication:

Figure 1: The original Figure 1 has been moved to the Supplementary Materials (Appendix). This decision was made to improve the narrative flow of the main text, The key interpretations from these figures are fully described in the text.

Table 2: To prevent the table from being visually overloaded and to focus on the most critical comparative metrics, we have removed the columns labeled "No Subsequent Complication" and "Subsequent Complication" (which contained the raw median values with interquartile ranges). The essential comparative information—the p-values, FDR-adjusted p-values, and the Ratio of Medians (MR/HR)—is retained and now presented more clearly. This streamlines the table while preserving all necessary statistical evidence for the identified markers.

Thank you again for your constructive comments, which have undoubtedly improved the quality and clarity of our manuscript.

Reviewer 2 Report

Comments and Suggestions for Authors

The authors did all required recommendations. I appreciate their responses. The paper could be published in the current form.

Author Response

Dear Reviewer,

Thank you very much for your positive final assessment and for your valuable time and insightful comments throughout the review process. We are very pleased that you find the manuscript suitable for publication in its current form.

Your guidance has been instrumental in improving the quality of our work.

Reviewer 3 Report

Comments and Suggestions for Authors

The authors have clearly addressed the comments, and I have no further concerns.

Author Response

(The authors gave the same response as above.)
